# Indonesian Stakeholders’ Perspectives on Warning Signs and Beliefs about Suicide

**DOI:** 10.3390/bs14040295

**Published:** 2024-04-03

**Authors:** Diana Setiyawati, Nabila Puspakesuma, Wulan Nur Jatmika, Erminia Colucci

**Affiliations:** 1Center for Public Mental Health, Faculty of Psychology, Universitas Gadjah Mada, Yogyakarta 55281, Indonesia; wulanjatmika@gmail.com; 2Faculty of Medicine, Public Health, and Nursing, Universitas Gadjah Mada, Yogyakarta 55281, Indonesia; nabila.p@ugm.ac.id; 3Department of Psychology, School of Science and Technology, Middlesex University, London NW4 4BT, UK; e.colucci@mdx.ac.uk

**Keywords:** suicide prevention, Indonesia, lived experience, warning signs, cultural belief, qualitative research

## Abstract

The suicide rate in Indonesia is considered low among Asian countries, but the underreporting rate is at a staggering 303%, and the latest reports suggest an increase in suicidal behaviour, particularly among young people. As a multicultural country, Indonesia has a complex system of beliefs about suicide. Thus, various aspects specific to Indonesia must be considered in understanding and preventing suicide. This paper explores Indonesian stakeholders’ perspectives through semi-structured interviews and focus group discussions. A total of 9 participants were individually interviewed, and 42 were involved in focus group discussions. They were mainly people with lived experiences of suicide. The other stakeholders were Indonesian experts who have experience in dealing with suicidal behaviour, helping people with a lived experience of suicide, or were involved in suicide prevention. Indonesian stakeholders highlighted various general and contextualised aspects concerning suicide. These aspects included a wide range of cultural beliefs and culturally specific warning signs, which included “*bingung*” (confusion) and longing for deceased persons. Other cultural beliefs such as viewing suicide as infectious, unpreventable, and guided by ancient spirits, and as an honourable act in some circumstances, also emerged. These findings can inform suicide prevention programs, including suicide prevention guidelines for Indonesia.

## 1. Introduction

Suicide is a devastating problem worldwide. According to the World Health Organization [1], approximately more than 703,000 people are reported to die by suicide each year, of which 77% of cases occur in low- and middle-income countries (LMICs), including Indonesia. However, in Indonesia, precise statistics of the suicide rate remain uncertain. Estimates suggest a range of 0.71–3.7 cases of suicide per 100,000 persons each year [2,3,4,5,6]. As the Indonesian population at the end of 2021 reached 273,800,000 people, these estimates equal to 1900–10,000 cases annually. In 2019, the WHO [3] estimated 6544 suicide cases in Indonesia, placing Indonesia in the top 15 countries with the highest number of suicides. Cases of suicide are found in every province in Indonesia, with provinces on the highest-populated Java Island accounting for 35% of all cases [5,6]. Papua, North Sumatra, and Bali are other areas with a higher prevalence [5,6]. The Indonesian Police Force has been documenting suicide reports since 2018 and showed an upward trend in the numbers over the years. However, the numbers might not reflect the true magnitude of suicide in Indonesia, since according to the Indonesian Association for Suicide Prevention, the unreported suicide rate in Indonesia is at least 303.6% [7]. Recent data show a particularly high prevalence of suicidal behaviours among young people, with approximately 5% of them reporting suicide attempts [8].

The COVID-19 pandemic appears to have impacted the mental health of people across the world, including those with suicide-related conditions. After the pandemic, efforts were made to study suicide trends in countries during and post-pandemic, resulting in mixed findings. During the early months of the pandemic (April–July 2020), a study using interrupted time-series analysis of 21 high-income and upper-middle-income countries observed no increase in suicide cases in all countries, yet found a decreased number of suicides in 12 countries [9]. A recent systematic review of the COVID-19 impacts on suicide attempts and rates found that out of 18 articles included, 4 articles (22.2%) reported increasing trends of suicide attempts, while 2 (11.1%) articles reported decreasing attempts, and 1 (5.6%) article reported no change [10]. While none of the studies reported in the review were conducted in Indonesia, police headquarters recorded a 30.4% increase in suicide cases between 2021 and 2022 in this country [11,12].

There is a wide range of factors associated with suicide risk. A meta-analysis of psychological autopsy studies has revealed that the biggest risk factor for suicide in adults aged 18–65 is mental disorders (OR = 13.1; 95% CI: 9.9 to 17.4), with depression being the biggest risk among other mental disorders (OR = 11.0; 95% CI: 7.3 to 16.5) [13]. A history of self-harm (OR = 10.1; 95% CI: 6.6 to 15.6) and suicide attempt (OR = 8.5; 95% CI: 5.3 to 13.4) are also associated with suicide rates. Other risk factors include sociodemographic factors (i.e., social isolation, unemployment, and low socioeconomic status) and a family history of mental disorders, suicide, and attempted suicide. Moreover, adverse life events (e.g., relationship conflict, legal problems, and family-related conflicts) are also a risk factor for suicide, with events happening during the past month having the biggest association with suicide (OR = 10.4; 95% CI: 7.1 to 15.3). In Indonesia, suicide affects all gender and age groups. Suicide deaths in males are three times higher than in females. The elderly population have the highest rates of suicide (18.2 per 100,000 persons), while those aged 25–49 have the highest number of cases, accounting for 46% of all cases [4]. However, contrary to characteristics in high-income countries, most people who die by suicide in Indonesia do not have mental health diagnoses: in Indonesia, 71% of suicide victims did not have a physical or mental illness when they died, while 23.2% reported having a mental illness, and 5.8% had a chronic physical illness [4].

Along with these factors, suicide is also influenced by sociocultural elements. According to Colucci and Lester [14], suicide has different meanings in every culture and is shaped by belief systems. Culture may influence suicidal behaviour in various ways, such as in methods of choice, general attitudes, and seeking help [14,15]. Similarly, the Culture Model of Suicide proposes three principles that correspond to this argument, namely that (1) culture may influence the types of stressors that result in suicide; (2) cultural meanings may impact suicidal tendencies, tolerance for psychological pain, and the following suicidal acts; and (3) suicidal intentions, thoughts, plans, and behaviour may present differently depending on the culture [16]. It has often been implied that every society also has a unique belief or conception of death, dying, and suicide. For instance, in Japan, attitudes regarding suicide have been historically influenced by the philosophy of upholding honour in death rather than continuing life in shame due to degenerated social roles [17]. In Indonesia, a myth called *pulung gantung*, which takes the shape of a meteor or fireball, is said to be a precursor of suicide by some people who live in Gunung Kidul, a region in Central Java [18]. Variations in attitudes towards suicide have also been discovered among youths from different cultures, with Indian young people having the most negative attitudes compared to those living in Italy and Australia [14].

Such examples underscore how belief systems need to be understood in order to develop culturally adequate and effective public health interventions for suicide. Culturally sensitive mental health interventions have shown effectiveness in reducing symptoms and might increase adherence to treatment [19]. In 2021, the World Health Organization released an implementation guide for suicide prevention that further stresses the significance of acknowledging, understanding, and considering cultural and religious beliefs in designing and implementing suicide interventions [3]. Studies have suggested the role of cultural and religious elements both as risk and protective factors for suicide, and understanding the different cultural contexts of suicide is important as a basis for suicide prevention and to increase its acceptability in the target populations [20,21,22].

A culture-based approach to suicide interventions often harnesses the leverage of the community. In the US, for example, participation in the Nia Project—a culture-based, empowerment-based group intervention for abused, low-income, and suicidal African-American women—was associated with a reduction in suicidal ideation and symptoms of depression compared to involvement in standard care [23]. A systematic review by Clifford, Doran, and Tsey [24] revealed that culturally informed suicide prevention initiatives have shown some positive outcomes in the form of decreased suicide probability, increased problem-solving skills, and increased protective behaviours against suicide among young people.

Considering the significance of local values and the sociocultural–political context of suicide to ensure the applicability and appropriateness of suicide prevention and interventions, paired with the scarcity of such publications, this research study aims to investigate the views of relevant stakeholders from Indonesia. More specifically, it seeks to gain insights into the warning signs, cultural beliefs, and perspectives on how Indonesian lay people should help suicidal persons, which, to the authors’ knowledge, have not been previously addressed in this country; therefore, this study will contribute to increasing the understanding and development of adequate suicide prevention strategies for Indonesia. This qualitative exploration is part of a larger mixed-method project aimed at co-developing suicide prevention guidelines for Indonesia, which are available at http://ugm.id/spgi (accessed on 31 January 2024) [25].

## 2. Materials and Methods

### 2.1. Design

To understand Indonesian stakeholders’ perspectives on the warning signs and cultural beliefs, focus group discussions (FGDs) and semi-structured in-depth interviews were conducted.

### 2.2. Participants

Nine participants were involved in the semi-structured interviews. They were people with lived experience of suicide (i.e., they tried to kill themselves or had a clear plan to kill themselves and/or someone close to them tried to kill themselves or died by suicide). The focus group discussion involved 42 Indonesian experts who had experience in helping people with a lived experience of suicidal behaviour and were involved in suicide prevention, such as government, members suicide book writers, community and religious leaders, psychiatrists, psychologists, nurses, and police officers. Participants from consumer groups who had a lived experience of suicide were also involved in focus group discussions. Table 1 presents the characteristics of the participants.

Participants were invited through the following methods:Snowball sampling: The stakeholders were identified through the network of the Center for Public Mental Health, Faculty of Psychology, Universitas Gadjah Mada (Yogyakarta, Java). From the experts of those networks, researchers asked for other experts in the field who were also invited.Publication: the researchers reviewed the Indonesian literature on suicide, identified authors’ names, and invited them to take part either in the semi-structured interviews or FGDs.

### 2.3. Procedure

A formal invitation letter was sent to participants through various methods, such as email, fax, post, or direct visit, depending on the specific circumstances. A plain-language statement and consent form were handed out to the participants prior to the semi-structured interview and FGD. The location for the interview was decided together with the participant. The focus group discussions were conducted at three different locations, i.e., Yogyakarta, Jakarta, and Gunung Kidul. The facilitators of the groups were SF, DS, NP, FH, and EC. All of the focus group discussions and semi-structured interviews were voice-recorded, and they lasted approximately 60–90 min. The structure of the semi-structured interviews and focus group discussions was developed based on the research questions.

### 2.4. Data Analysis

Thematic analysis of the transcripts was employed to analyse the data following the method of Clarke and Braun [26] by three researchers D.S., N.P., W.N.J. under the supervision of E.C. The data were processed through NVIVO 12 software. Emerging similar nodes were highlighted and categorised into two superordinate themes and thirteen subordinate themes.

## 3. Results

The results are presented as follows, accompanied by example quotes from the participants. The numerical code represents individual participants, while “INT” indicates a quote from an individual in-depth interview, and “FGD” is from the focus groups.

### 3.1. Suicide Warning Signs

In addition to those mentioned in the global literature (e.g., hopelessness and no sense of purpose in life), participants identified specific warning signs of suicide among Indonesians. These ranged from expressions of fear, confusion (expressed as “*bingung*” by participants), and longing for a deceased person (relative).

#### 3.1.1. “Bingung”

Participants mentioned “*bingung*” as an obvious warning sign that occurred within a week before the person took their own life. “*Bingung*” is an Indonesian word that literally means “confused”. The participants defined the term as restlessness, unhappiness with being at home, and even the act of running away from home (e.g., to a farm field located far from the village or walking a considerable distance without specific direction). One participant described his father before he took his life as follows: “A bit stressed, “*bingung*”” (INT-2). Another participant described that “Before taking his own life, he looked “*bingung*”, then ran away to the farm field. People found him lying down on the ground and looking “*bingung*” (INT-3). One participant shared the story and remarks made by a relative before he decided to attempt suicide: “I feel ‘*bingung’*, where is my house, I feel like I am not the son of my father and mother” (INT-15).

#### 3.1.2. “Ajrih”

“*Ajrih*” is a Javanese word, referring to being in a state of fearfulness. Participants described “*ajrih*” as an intense fear of something. “My grandfather showed intense fear sometimes before he took his own life.” (INT-4). Another participant explained that her father had an intense fear of flags before taking his own life. “One week before, he was ‘*ajrih*’. If he saw a flag in the neighbourhood, he would be afraid of it. He felt fear during the daytime. The flag of a political party”.

#### 3.1.3. Longing for a Deceased Person (Usually a Relative)

Participants also mentioned that the person who took their life conveyed a longing to meet a deceased person. A participant referred to his grandfather as follows: “He wanted to meet his brother, wanted to send flowers to his graveyard…” (INT-4). This belief is related to the Islamic faith belief that there is an afterlife or that the soul is eternal.

#### 3.1.4. Changing Identity

Participants mentioned that the person who died by suicide wanted to change their name or identity beforehand:

“*One week before taking his own life, he came to my house and asked me to change his name. He brought along his elementary school certificate with black blocked ink on top of his name and asked me to change it (his name). He looked ‘bingung’*”(INT-8)

This behaviour is also closely related to the Javanese tradition of changing names if someone is ill or living in unfortunate conditions.

#### 3.1.5. Writing a Sad Story

Some participants observed writing sad stories related to death as a sign of making a suicidal plan, for example, “So writing a story, where in the story the protagonist died by suicide” (S-11). “My experience was I wrote a story in the magazine, where the main character died. Or poetry about death. Not always by suicide; it can be by accident or illness. In the most important plot, the main character died” (FGD-12).

#### 3.1.6. Withdrawal

Several participants mentioned that social withdrawal is a very typical warning sign, for example, “Usually if the suicide plan was already clear, he/she would withdraw from friends and family” (FGD-13). Another participant elaborated that “Yeah, that is very typical; for sure, he/she doesn’t want to see people and does not want to do things. Doing self-isolation, because if going out he/she thinks that people will look at him/her and knows his/her problems” (FGD-12). This participant also gave a clear example of his direct experience: “Actually, I also got bored, always staying in my room. I only went out at night for food and would quickly return to avoid people” (FGD-12).

#### 3.1.7. Preference to Be in a Dark Room

Natural and artificial light was also indicated as a source of discomfort for some people with suicidal thoughts; for example, one participant said that “I wanted to be in a room that is not always bright” (FGD-14). Another participant highlighted that “I didn’t want to be in any room that is bright” (FGD-15). The light also includes the sun, as one participant said “If the room had a curtain, I would always close it. I only went out at night, around 11 pm.” (FGD-14).

#### 3.1.8. Differences in Methods among Males and Females

A few participants also noticed some gender differences that might impact the warning signs shown by males and females, for example, “I noticed the differences. If a man planned to take his own life, the action would be very concrete and straightforward, such as buying a hanging rope or pesticide or going straight away to a railway line. If a woman had a plan, she would still think of peaceful ways to die or find a way to die without feeling pain” (FGD-11). Another participant remarked that “Since women like to think a lot, the possibility of being helped is big” (FGD-12). Similarly, another female participant also said that “I would be in my room a lot, thinking of how to die comfortably” (FGD-13).

### 3.2. Beliefs about Suicide

#### 3.2.1. Suicide Is an Honourable Act

In Indonesia, becoming a burden for other people is considered shameful, more shameful than taking one’s own life. Giving social support, such as visiting sick people, giving money, or helping with basic needs, is common in Indonesian society. For the sick person, social support is considered a burden because they feel obligated to return that almsgiving. Therefore, for some, it is better to take one’s own life than be a burden for the neighbours or relatives, as mentioned by this participant: “There is belief in our society that dying by suicide is more honourable than living as a burden for others” (FGD-20). Taking one’s own life is also considered a courageous action, as highlighted by another participant: “The society will think that they are fighters, so they are not afraid of taking their own life” (FGD-21).

#### 3.2.2. Suicide Is Guided by Ancient Spirits

There is a belief in Indonesia that suicide is often influenced by ancient spirits, as highlighted by one participant: “One person who tried to take his own life reported that he threw himself into an old well because he was guided by an ancient spirit” (FGD-21). Another participant mentioned that his neighbour was guided by someone unseen to take his own life: “he was being guided by unseen voice” (S-13).

#### 3.2.3. Suicide Is Infectious

Another widespread belief is that suicide is infectious, as mentioned by the following participant about his father-in-law: “We asked him, why do you want to send flowers to your brother’s graveyard, he died by suicide, suicide is infectious” (INT-5).

#### 3.2.4. “Pulung Gantung” Is an Early Sign of Suicide

While most participants with higher education suggested that suicide should never be the outcome, they also thought that sometimes life stressors around the victims could be too much to bear. Other participants, especially those who resided in Gunung Kidul, a sub-district in the Special Region of Yogyakarta, firmly believed that suicide (particularly by hanging) is simply part of God’s will (called “*pulung gantung*” (*Pulung gantung* is a local myth in Gunung Kidul. According to this myth, if a flying meteor-like fireball falls upon a village, it signifies that someone in that village will die by hanging. “*Pulung*” is a Javanese word that translates to a gift or a sign, while "*gantung*" means hanging.) in Javanese). This is expressed by a quoted example here from a participant: “*Pulung gantung*” is, like, when we sit here and see fire (like a shooting star) flying to that house, it means that someone in that house has been preordained to die by suicide” (INT-15).

#### 3.2.5. Suicide Is Not Preventable

Beliefs such as those indicated above led a few participants to express disbelief that suicide can be prevented: “What? The government wants to make a suicide prevention? It is impossible. If someone is chosen to die by suicide, he will not be able to hide. Somebody, like, guided him to do this, like my neighbour. I heard he talked to somebody before committing suicide. We could not see to whom he talked to… so suicide could not be prevented” (FGD-13).

## 4. Discussion

Life and death, including death by suicide, are inherently universal concepts, yet they also exhibit cultural specificity, as also argued by Chu and collaborators [16] and Colucci [27]. This concept also applies to suicide warning signs [27]. Many of the warning signs found in this research study are universal, while some are culturally bound. An example of a universal warning sign is ‘*ajrih*’ or intense fear. Even though it is a specific term popular in the Javanese community, this term may be a manifestation of anxiety. Anxiety is one of the warning signs agreed upon by expert consensus [28]. Hopelessness is another universal warning sign implied in this study, for example, by writing a sad story. Withdrawal, another universal warning sign [28], is expressed in this research study by participants as a preference to “be in a dark room” and avoid meeting people during the daytime.

What is also universal is that women attempt suicide more frequently, while men, though attempting less often, experience higher mortality rates due to the use of more lethal means. This finding is in line with Trisigotis et al.’s (2011) study which claimed that women attempted suicide more frequently compared to men, who chose more lethal suicide methods. In Europe, women have demonstrated more attempts at suicide but fewer completed suicides compared to men [29].

Some of the universal and specific warning signs that we found in this study are congruent with the interpersonal theory of suicide that posits that thwarted belongingness together with perceived burdensomeness creates the desire to commit suicide, and once someone has the capabilities to act out the suicide, the risk of suicide becomes lethal [30]. The thwarted belongingness in this study was manifested by ‘*ajrih*’, preference to “be in a dark room”, and avoiding meeting people. Meanwhile, the perceived burdensomeness was also present as the person who died by suicide shared the thought of becoming a burden if they continued to live.

One culturally specific warning sign that emerged in this study is “*bingung*”, a typical Javanese expression for confusion, not knowing what to do or where to go [31]. The term is also commonly used to express identity confusion, as mentioned by the participant who indicated that before attempting suicide, they wanted to change their identity. The participant used the word “*bingung*” to refer to a sense of identity, which is connected to the concept of “*kabotan jeneng*”, a Javanese concept that can be translated as “burdened by the name”. When someone feels like their name has too high a spiritual burden, usually manifested in a severe illness or misfortune, they are advised to change their name [32].

Another culturally specific warning sign is the expression of longing for deceased relatives. By conveying the desire to follow the departed and join them, the victims express an implicit longing for their own passing. Such implicitness in communication is commonly applied in Javanese society [33].

Some differences that might be reflected in the warning signs shown by individuals have also been suggested. In regard to suicide beliefs, we observed some similarities with the existing literature in other contexts. The association of the act of suicide with honour has been observed also in other societies. For example, female suicide cases in China are associated with the notion of preferring death over living without meaning [34]. This phenomenon bears resemblance to the historical belief of suicide in Japan, which values death above feelings of shame and views taking one’s life as a courageous act, as understood in the practice of *seppuku* [18]. According to Durkheim’s framework, such an idea reflects the concept of altruistic suicide, where suicide correlates with a high level of integration and expectation to succeed in society [35].

Another culturally bound belief indicated by participants connects suicide with ancient spirits, although the idea of suicide being guided by some spiritual or incorporeal existence is shared by other cultures as well. Suicide survivors in Yucatan, Mexico, often describe their suicidal attempts as mystical instances, with voices and a demonic presence reported to have appeared and ‘tempted’ them before the attempts [36]. In another part of Mexico, the Chol Mayan indigenous people also regard witchcraft as a cause of suicide [37]. The society of Akan in Ghana echoes similar sentiments by linking the cause of suicide to supernatural forces, including sorcery [38]. However, to the best of our knowledge, this is the first published study in Southeast Asia where such a belief has been reported.

Some participants in this study also believe that suicide is infectious and that it is a form of a communicable disease spread by an external agent. In a systematic review of the exploration of the concept of suicide contagion conducted by Cheng et al. [39], several papers refer to ‘contagion’ in the context of ‘transmission’. However, it is used as a metaphor for ‘transmitting’ cognitive elements, such as information on lethal suicide methods, impulsive or aggressive behaviours, or the disposition to perform the acts. The way in which the Indonesian participants view suicide as something infectious also holds different meanings from other contexts of contagion, such as imitation (referring to conformity) or affiliation (referring to homophily) [39], making it a unique belief.

Reflecting on the existing theory of suicide, some of the universal and specific warning signs that we found in this study align with the interpersonal theory of suicide. This theory posits that thwarted belongingness, coupled with perceived burdensomeness, is significantly associated with a desire for suicide. Moreover, once individuals possess the capabilities to carry out suicide, the risk becomes lethal [40]. In this study, thwarted belongingness was manifested by ‘*ajrih*’, a preference for being in a dark room, and avoiding social interaction. Meanwhile, perceived burdensomeness was also present, as the person who died by suicide expressed thoughts of becoming a burden if they continued to live.

Finally, although this project largely aimed to elicit Indonesian-specific items to be incorporated in a Delphi study used to develop a suicide prevention strategy, namely, the Suicide prevention first aid guidelines for Indonesia [26], some participants indicated a lack of belief that suicide is at all preventable. Previous studies showed that there is a variety of perspectives in culturally diverse communities about whether or not suicide can be prevented. For instance, Colucci [28] noted that more students in India believe that suicide is not preventable compared to students in Australia and Italy. On the other hand, in a study conducted by Hjelmeland et al. [41], students in Ghana, Uganda, and Norway mostly agreed that suicide could be prevented. One probable explanation for the pessimistic attitude towards suicide preventability found in our study might stem from the commonly accepted belief among Indonesians that everything is predetermined in life. Most Javanese adopt the concept of *nerimo,* which essentially translates into the recognition that one cannot escape from the inevitability of destiny, including matters related to death [42]. This belief seemed to be particularly strong in Gunung Kidul. This part of Indonesia has not only reported higher rates of suicide compared to others (4.48 persons for every 100,000), but it is also characterised by a special form of suicide, *pulung gantung*, which was discussed above in the data collection explanation. Such beliefs need to be taken into consideration when developing suicide prevention strategies for Indonesia.

This research study has contributed valuable insights into local beliefs and specific warning signs in the context of Indonesia, with a specific focus on Javanese culture. The collection of views expressed by a wide range of stakeholders, many of whom were people with a lived experience of suicide, enriches the existing literature on warning signs and suicide beliefs by highlighting both universal aspects and those with deeply rooted local significance and expression.

One of the limitations is that data collection was conducted only in some parts of Java. However, these qualitative findings were integrated into a three-round mixed-method Delphi study that recruited experts/participants across Indonesia [26].

## 5. Conclusions

This is the first qualitative study in Indonesia that has explored stakeholders’ views, including people with a lived experience of suicidal behaviour, regarding warning signs for suicide and beliefs about suicide and suicide prevention in Indonesia. As emphasised by the culturally specific beliefs that emerged in this study, it is crucial to explore the sociocultural context of suicide and suicide prevention (including cultural meanings and myths, such as the ‘shooting star’) instead of adopting what has been developed in other countries. Suicide research is currently insufficient in Indonesia, lacking accurate data on suicide attempts and mortality. While our study contributes potentially valuable insights into beliefs about suicide, larger and more in-depth studies across Indonesia are needed to guide the national suicide prevention efforts [4].

## Figures and Tables

**Table 1 behavsci-14-00295-t001:** Characteristics of participants.

Attribute	Frequency	Percentage
Interview		
Gender		
Male	7	77.78
Female	2	22.22
Profession/Role		
Person with lived experience	9	100
Focus Group Discussion		
Gender		
Male	16	48.48
Female	17	51.52
Profession/Role		
Psychologist working in primary health care	5	15.15
Psychiatrist	3	9.09
Policy maker	3	9.09
Health officer	2	6.06
Mental health cadre	6	18.18
Author of books about suicide	2	6.06
Community leader	5	15.15
Religious leader	2	6.06
Health cadre	1	3.03
Cultural expert	1	3.03
Paediatrician	1	3.03
Public health practitioner	2	6.06

## Data Availability

Data are unavailable due to privacy reasons.

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
