# Peer review of "Indonesian Stakeholders’ Perspectives on Warning Signs and Beliefs about Suicide"

_behavsci, 2024, doi:10.3390/bs14040295_

Round 1

Reviewer 1 Report

Comments and Suggestions for Authors

Thank you for the interesting and well-written review of the international literature and comparison to current experiences in Indonesia in the Introduction. The choice of semi-structured in-depth interviews was appropriate to the research intent.

The finding of the theme of “Bingung” in 3.1 suicide warning signs is very interesting, and a unique new insight into cultural differences in suicide presentations, and therefore relevant to design interventions required. The theme of “Ajrih” is also interesting and potentially warrants further research in the future to separate suicidaility in the general population and in those experiencing active psychosis. The data in 3.2 is very interesting to understand stigma in Indonesia in relation to suicidal behaviour.

The Discussion includes fascinating meaning-making around similarities and differences, due to cultural differences, in the international consensus about suicide risk indicators. The notion of 'altruistic suicide' is well discussed and deserves a greater exploration in future research studies, internationally.

The Conclusion is solid and need not commence with the words 'To the best of our knowledge,' page 9, line 364.

Comments on the Quality of English Language

There are minor but important edits required to strengthen expression.

The word 'solid' page 1 line 16 in the Abstract is unclear in meaning.

Also in the Abstract, if people are participation in providing data they do not have an experience of suicide, they have an experience of attempted suicide; please amend, page 1 line 21. This issue is also to be addressed in Method page 3 line 133; perhaps the word 'suicidality' would be better? "a lived experience of suicidality"

Page 6 line 225, a better word than 'inconvenience' here would be discomfort.

Author Response

Dear reviewer, thank you very much for your positive feedback. Below are the revisions we made according to your suggestions:
1. We have deleted the word 'To the best of our knowledge,' page 9, line 370.
2. We have deleted the word 'solid', page 1 line 16.
3. The word 'lived experience of suicide' has been explained on page 3 lines 133-134.
4. Thank you for your suggestion. We have changed the word 'inconvenience' to 'discomfort', page 6 line 225.

Reviewer 2 Report

Comments and Suggestions for Authors

This is a great contribution to the literature and there are several strengths that you might consider highlighting. This study, even with a relatively small sample, incorporates perspectives from a diverse group. The proportion of men is noteworthy in comparison to many other studies. Additionally, recruiting individuals with lived experience in combination with professionals gives the data more breadth than might otherwise be achieved. There are a few considerations for the analysis.

For the concept of Ajrih, or fearfulness, although it parallels anxiety there is a contrast with Joiner's Interpersonal Theory concept of acquired capability (associated with fearlessness). It is thus important to distinguish what it is that the individuals are fearful of or if it is a general state of anxiety (i.e., feeling 'on edge).

For the concept of suicide as an honourable act, it may again be useful to consider how the results align (or not) with the Interpersonal Theory concept of perceived burdensomeness. Other theories point to the importance of intense feelings of shame. In this study, it may be important to distinguish whether the key component is related to self-judgment (burden) or ideas about the perceptions of others (shame).

For the guidance of ancient spirits, in the context of psychiatric symptomatology, it may be useful to distinguish between spiritual experience and auditory hallucinations (if possible).

The final suggestion is a minor point. Take care where your citations appear. For reference 7, it seems that the intent is for it to support the contention of under-reporting. However, the placement of the citation (lines 45-47) might seem like a reference for the Indonesian Association for Suicide Prevention as an organization. Placing the citation at the end of the sentence will make it more clear that it is intended to support the concept and not the organization.

Author Response

Dear reviewer,

  Thank you very much for you positive feedback and suggestions. Please find our answer to your feedback below:   1. Reviewer's comment: For the concept of "ajrih", or fearfulness, although it parallels anxiety there is a contrast with Joiner's Interpersonal Theory concept of acquired capability (associated with fearlessness). It is thus important to distinguish what it is that the individuals are fearful of or if it is a general state of anxiety (i.e., feeling 'on edge).   For the concept of suicide as an honourable act, it may again be useful to consider how the results align (or not) with the Interpersonal Theory concept of perceived burdensomeness. Other theories point to the importance of intense feelings of shame. In this study, it may be important to distinguish whether the key component is related to self-judgment (burden) or ideas about the perceptions of others (shame).

Answer:
We have added a paragraph about the reflection on the interpersonal theory of suicide on page 8 line 334.   2. Reviewer's comment: For the guidance of ancient spirits, in the context of psychiatric symptomatology, it may be useful to distinguish between spiritual experience and auditory hallucinations (if possible).   Answer: Unfortunately, we cannot make sure of this because we got the data from the relatives of the person who died by suicide.   3. Reviewer's comment:
The final suggestion is a minor point. Take care where your citations appear. For reference 7, it seems that the intent is for it to support the contention of under-reporting. However, the placement of the citation (lines 45-47) might seem like a reference for the Indonesian Association for Suicide Prevention as an organization. Placing the citation at the end of the sentence will make it more clear that it is intended to support the concept and not the organization.
  Answer: Thank you for your suggestion, we have moved the citation at the end of the sentence.